

# CRISPR screening identifies M1AP as a new MYC regulator with a promoter-reporter system

Akiko Yamamoto[*], Morito Kurata[*], Iichiroh Onishi, Keisuke Sugita, Miwa Matsumura, Sachiko Ishibashi, Masumi Ikeda, Kouhei Yamamoto and Masanobu Kitagawa

Department of Comprehensive Pathology, Graduate School of Medical and Dental Sciences, Tokyo Medical and Dental University, Tokyo, Japan
[*] These authors contributed equally to this work.

## ABSTRACT

**Background**. *MYC* is one of the proto-oncogenes contributing to tumorigenesis in many human cancers. Although the mechanism of *MYC* regulation is still not fully understood, learning about the comprehensive mechanism controlling the transcriptional activity of *MYC* will lead to therapeutic targets. The CRISPR/Cas9 library system is a simple and powerful screening technique. This study aims to identify new transcriptional upstream activators of *MYC* using the CRISPR activation library with new promoter-reporter systems.

**Methods and Results**. The *MYC* promoter-reporter system was developed with a photoconvertible fluorescent protein, Dendra2, and named "p*MYC*-promoter-Dendra2." This *MYC* promoter-reporter system was designed to harbor a proximal *MYC* promoter at (3.1 kb). Both the CRISPR activation library and p*MYC*-promoter-Dendra2 were induced to HEK 293T cells, and Dendra2-positive cells, that are supposed that *MYC* should be upregulated, were collected individually by a cell sorter. Among the 169 cells collected, 12 clones were successfully established. Then, p*MYC*-promoter-Dendra2 was transfected again into these 12 clones, and two of 12 clones showed Dendra2 positivity. In this procedure, the cells with non-specific autofluorescence were correctly distinguished by utilizing the photoswitchable character of Dendra2. Using extracted genomic DNA of these two Dendra2 positive clones, polymerase chain reaction (PCR) was performed to amplify the guide RNA (gRNA) containing region, which was introduced by the CRISPR activation library. Eventually, *PLEKHO2*, *MICU*, *MBTPS1*, and *M1AP* were identified, and these gRNAs were transfected individually into HEK 293T cells again using the CRISPR activation system. Only *M1AP* gRNA transfected cells showed Dendra2-positive fluorescence. Then, the overexpression vector for *M1AP* with a doxycycline-inducible vector confirmed that *M1AP* induced high *MYC* expression by real-time quantitative PCR and western blot. Furthermore, the dual-luciferase assay showed a significant increase of promoter activity, and *MYC* mRNA was higher in *M1AP*- overexpressing cells. *M1AP* is highly expressed in several cancers, though, a positive correlation between *M1AP* and *MYC* was observed only in human acute myeloid leukemia.

**Conclusion**. The present study confirmed that the experimental method using the CRISPR library technology functions effectively for the identification of molecules that

Corresponding author
Morito Kurata,
kurata.pth2@tmd.ac.jp

activate endogenous *MYC*. This method will help elucidate the regulatory mechanism of *MYC* expression, as well as supporting further drug research against malignant tumors.

## INTRODUCTION

*MYC* is one of the most well-known proto-oncogenes, contributing to tumorigenesis in many human cancers. It can function as a transcriptional factor in both cancerous and non-cancerous cells, and it is involved in cell growth, proliferation, metabolism, and transformation (*Dang, 2012*). The Wnt- β-catenin signaling pathway is one of the most well-described pathways involved in the regulation of *MYC*. Due to mutations in the *adenomatous polyposis coli* (*APC*) gene or abnormalities in *glycogen synthase kinase-3* β, Wnt-signal is stimulated and the excessive accumulation of β-catenin occurs. Then, active β-catenin moves into the nucleus and binds to T cell factor (TCF), followed by inducing *MYC* expression (*Liu et al., 2002*; *Novak & Dedhar, 1999*). Mitogen-activated protein kinase (MAPK), which is a serine/threonine kinase, and its subfamily extracellular signal-regulated kinase (ERK), regulate *MYC* transcriptional activity by inducing phosphorylation of MYC (MAPK/ERK signaling pathway). Furthermore, as non-coding RNA, microRNAs like *miR-34* (*Kong et al., 2008*) and *let-7* (*Sampson et al., 2007*) suppress *MYC* expression by binding to the 3′-untranslated region of *MYC* mRNA. As for long noncoding RNAs, *PVT1* regulates the expression of *MYC* by inhibiting the degradation of MYC protein (*Tseng et al., 2014*). Thus, it has been reported that various regulatory mechanisms function in a complex fashion regarding *MYC* expression, although other molecules and elements like crosstalk between each pathway are not fully understood.

Concerning the high *MYC* expression rate of multiple malignant tumors, elucidating the mechanism controlling the transcriptional activity of *MYC* is a crucial issue because it will lead to therapeutic targets. Using the CRISPR/Cas9 library system, with its collection of more than 70,290 different guide RNAs (gRNAs), enables us to identify molecules that relate knockout or activation of endogenous genes. To date, the CRISPR library has been utilized for various studies. *Shalem et al. (2014)* identified genes whose loss is involved in resistance to vemurafenib by introducing the knockout library into a human melanoma cell line followed by treatment with *BRAF* inhibitor vemurafenib. In contrast, *Konermann et al. (2015)* identified molecules that exhibit resistance to PLX-4720 by introducing an activation library. In this way, the CRISPR library has been used to identify the certain genes involved in drug sensitivity or resistance, toxin resistance, and hypoxia response (*Konermann et al., 2015*; *Gilbert et al., 2014*; *Kampmann, 2018*; *Kurata et al., 2018a*; *Kurata et al., 2018b*; *Shalem et al., 2014*; *Wang et al., 2014*; *Koike-Yusa et al., 2014*; *Doench et al., 2016*; *Jain et al., 2016*); however, a screening of molecules that act on the promoter region of a specific gene and control transcriptional activity has not yet been tested. In addition, autofluorescence is highly problematic for collecting fluorescent cells in the screening.

Therefore, a photoconversion protein, Dendra2, was used in this system. Dendra2 is derived from the octocoral *Dendronephthya sp. T.* and can change color with UV-violet light (405 nm laser). This study aims to utilize the CRISPR activation library for the detection of new transcriptional activators of *MYC*. For this purpose, we developed an original tool for evaluating *MYC* activity and establishing screening methods.

## MATERIALS & METHODS

### *MYC* promoter-reporter system and lentivirus plasmids

The Polymerase chain reaction (PCR) primers were designed across the proximal *MYC* promoter at −3.1 kb from the MYC transcriptional start site (TSS), and PCR was performed with KOD-FX (Toyobo). The primers were 5′-TTAGCTAGCGAGGGTT-TTCTTTGAGGGGC-3′ (forward) and 5′-AACGGATCCCGGAGATTAGCGAGAGAGGA-3′ (reverse).

A plasmid of Dendra2 (pDendra2) was purchased from Takara-Clontech. The cytomegalovirus immediate early promoter (PCMVIE) was removed with restriction enzymes (Afl III and Nhe I), and the multi-cloning site was replaced with Nhe I and BamH I. Amplified fragments of −3.1 kb of the *MYC* promoter region with Nhe I and BamH I restriction enzyme sites were integrated into pDendra2.The new reporter system with Dendra2 and the −3.1 kb *MYC* promoter region (p*MYC*-promoter-Dendra2) was used for further experiments. The double-stranded DNA fragments of *M1AP* with attB sites were purchased from gBlocks® (IDT) and generated into the pENTR221 vector using Gateway® BP clonase. Then they were transferred to the TripZ-TRE-DEST-IRES-GFP-EF1A-rtTA vector (*Kurata et al., 2016*) using Gateway® LR clonase.

### Cell culture and SAM library screening

Human embryonic kidney cells (HEK 293T) were kindly provided by Dr. David Laraespada, Masonic Cancer Center, University of Minnesota, MN, USA. and maintained in Dulbecco's modified Eagle's medium (DMEM, Fuji Film) with 10% fetal bovine serum and penicillin/streptomycin with 5% $CO_2$ at 37 °C . To transduce the lentivirus, HEK 293T cells ($5 \times 10^5$ cells/well) were seeded in a six-well cell culture plate one day before the transfection. The next day, 3 μg of lentivirus plasmid was transfected with 1 μg of pMDG and 2 μg of pCMV using Lipofectamine™ 3000 Reagent (Invitrogen) according to previous work (*Kurata et al., 2016*). Twelve hours after transfection, the medium was changed to fresh DMEM. The virus supernatant was harvested at 48 h after transfection, then filtered with Millex™ -HP 0.45 μm (Millipore). The target HEK 293T cells ($5 \times 10^5$ cells/well) were seeded in a six-well plate one day prior to transduction and were transduced with this lentivirus supernatant with 5 μg/ml polybrene (Sigma). The used lentiviral plasmids were lentiMPHv2 (Addgene, #89308), lentidCAS-VP64_blast (Addgene, #61425), and lentiSAMv2 (Addgene, #61597). At first, lentiMPHv2 was transduced into HEK 293T and treated with Hygromycin B for 2 weeks. Then, lentidCAS9-VP64 was induced in MPH-expressing HEK 293T cells and treated with Blasticidin for 2 weeks. The MPH-dCas9-VP64-expressing HEK 293T applied to further experiments. The control and *MYC* activation gRNA (Fig. S1C) were induced in MPH-dCas9-VP64-expressing

HEK 293T cells. The CRISPR activation library, lentiSAMv2, was also transduced into MPH-dCas9-VP64-expressing HEK 293T cells and then treated with Zeocin (300 µg/ml, Invivogen) for 2 weeks.

## Collection of Dendra2-positive cells and identification of gRNAs

After 2 weeks of selection with Zeocin, p*MYC*-promoter-Dendra2 was transfected into SAM library lentivirus-transduced cells with Lipofectamine[TM] 3000 Reagent (Invitrogen). HEK 293T cells transfected with p*MYC*-promoter-Dendra2 were evaluated for Dendra2-positive cells by fluorescence microscope (Keyence). Cells were collected at 72 h and washed with phosphate-buffered saline (PBS) and applied to the cell sorter. The Dendra2-positive cells were collected into a 96-well plate with MoFlo XDP (Beckman Coulter).

A total of 169 cells were collected, and 12 clones were successfully established. Collected clones were centrifuged and washed with PBS solution and pelletized. Lysis buffer solution with proteinase K (20 mg/ml) was added to cells and left for 30 min at 60 °C. After treatment at 94 °C for 10 min, the lysate was centrifuged for 15 min, and the supernatant was collected as genome DNA (gDNA) containing fractions. Using extracted gDNA of each collected clone, PCR was conducted with GoTaq® qPCR Master Mix (Promega) with the following primers: 5′-GAGGGCCTATTTCCCATGAT -3′ (forward) and 5′-GAGGGCCTATTTCCCATGAT-3′ (reverse). PCR product containing the gRNA region was employed for TA cloning with pGEM®-T Vector Systems (Promega) and gRNAs were sequenced by Sanger-sequencing.

## Preparation of CRISPR backbone plasmid with target gRNAs and transcriptional activation for *MYC* using the CRISPR activation system

The oligonucleotides for each target gRNA sequence were designed using the list from the SAM library and integrated into the pENTR221/U6/stuffer/MS2 vector (*Kurata et al., 2018a*). The MPH-dCas9-VP64-expressing HEK 293T cells ($1 \times 10^5$ cells/ml) were seeded in a 24-well cell culture plate. The next day, 2.5 µg of individual gRNA vector and p*MYC*-promoter-Dendra2 were transfected with Lipofectamine [TM] 3000 Reagent. At 48 h after the transfection, the *MYC* transcriptional activation was evaluated by Dendra2-positivity by Keyence.

## Evaluation of *MYC* expression

A pTRIPZ-*M1AP-GFP* was transduced into the HEK 293T cells. These transduced cells were selected with puromycin (1 µg/ml, Invitrogen) for 2 weeks. Seventy-two hours after the induction of Doxycycline (DOX) (1 µg/ml), the collected cells were employed for RNA isolation.

RNA was isolated using an RNeasyR Mini Kit (Qiagen) according to the manufacturer's instructions. Complementary DNA (cDNA) was generated from RNA with ReverTra AceR qPCR RT Master Mix (Toyobo). *Beta-ACTIN* was used as an endogenous control. Using the ABI Prism 7900HT (Applied Biosystems), quantitative PCR (qPCR) analysis was performed to quantify the RNA level using SYBR Mix. The sequences for the PCR primers

used in gene expression were as follows: *MYC*: 5′-CGACTCTGAGGAGGAACAAGAA-3′ (forward) and 5′-CAGCAGAAGGTGATCCAGACT-3′ (reverse), β-*ACTIN*: 5′-CACAGAGCCTCGCCTTTGCC-3′ (forward) and 5′-CACAGAGCCTCGCCTTTGCC-3′ (reverse). The mRNA level of the targeted gene was analyzed by comparison with the standard calibration curve. Western blot analysis was performed according to standard procedures (*Kurata et al., 2016*). The primary antibodies used were anti-c-Myc (Abcam ab32072) and anti-GAPDH (Santa Cruz sc47724), each at a dilution of 1:1000. The primary antibodies were detected with horseradish peroxidase (HRP)-conjugated secondary antibodies at 1:50000. Protein bands were visualized with Clarity$^{TM}$ Western ECL Substrate (Bio-Rad).

## Dual-Luciferase reporter assay

The fragments of the *MYC* promoter was integrated into a pGL4 vector (Promega). A pT3.5-CAG-*M1AP* was generated with pENTR221-*M1AP* and pT3.5-CAG-DEST (*Kurata et al., 2018a*) using Gateway® LR clonase. The luciferase reporter vectors, the pT3.5-CAG-*M1AP* vector, and the pRL Renilla luciferase reporter vector were co-transfected in the HEK 293T cells. The samples were harvested 48 and 72 h after the transfection. Each assay was biologically triplicated and repeated. Luciferase activities were measured using the dual-luciferase reporter assay system (Promega) and Lumat LB9507 (Perkin Elmer). RNA was harvested at the same time points and the expression of *MYC* mRNA was analyzed by qPCR.

## *M1AP* and *MYC* expressions in databases

For the purpose of comparing gene expression between cancer and non-cancerous tissue, the GEPIA web server was employed (*Tang et al., 2017*) (http://gepia2.cancer-pku.cn/).

Datasets for AML samples were used in BloodSpot database, that database was used to retrieve gene expression data from curated human and murine microarray and RNA-sequencing (RNA-Seq) datasets (*Bagger et al., 2016*). GSE42519, GSE13159, GSE15434, GSE61804, GSE14468, and The Cancer Genome Atlas (TCGA) datasets were used for analysis. In addition, following database were used; The Human Protein Atlas (https://www.proteinatlas.org/) and cBioPortal (https://www.cbioportal.org) using TCGA data (*Cancer Genome Atlas Research Network et al., 2013*; *Gao et al., 2013*; *Cerami et al., 2012*).

## Statistical analysis

The data were statistically analyzed using the EZR version 1.36 software. A value of $p < 0.05$ was considered statistically significant for all analyses.

# RESULTS

## Development of the *MYC* promoter-reporter system for library screening

Using PCR primers designed across the proximal *MYC* promoter at −3.1 kb (−3134 to −40) from the *MYC* TSS, PCR was performed. The *MYC* promoter region was integrated upstream of *Dendra2*. The pDendra2 harboring the 3.1 kb *MYC* promoter region was

called "p*MYC*-promoter-Dendra2" for further experiments (Fig. S1A). Plasmids of p*MYC*-promoter-Dendra2 and gRNA of *MYC* or *HPRT* (as a negative control) were transfected in HEK 293T cells using the CRISPR activation system. We confirmed that Dendra2 showed positivity only in the cells with gRNA for *MYC* (Figs. S1B, S1C). Therefore, p*MYC*-promoter-Dendra2 was utilized as a *MYC* promoter-reporter analysis tool in this study.

### *MYC* transcriptional activator screening Using the CRISPR activation library

To screen transcriptional activators working on the *MYC* promoter region, the CRISPR activation library was employed. The SAM library lentivirus was infected into MPH-dCas9-VP64-expressing HEK 293T cells ($1 \times 10^7$ cells). After selection, p*MYC*- promoter-Dendra2 was transfected into these HEK 293T cells. The multiplicity of infection (MOI) ranged from 1.4 to 5.2.

At 72 h after transfection, cells were proceeded to Dendra2-positive cell collection using flow cytometry (positive ratio: 0.017%). Sorted single cells were collected into a 96-well plate and incubated for 2 weeks. Among the collected 169 cells, 12 clones were successfully established (Fig. 1).

### Second screening to identify candidate genes as *MYC* activators

For the second screening, p*MYC*-promoter-Dendra2 was again transfected into the 12 clones. Using fluorescence microscope, clones 3 and 5 were shown to be truly Dendra2 positive (Figs. 2A–2L), and the photoswitchable attribute of Dendra2 was useful for distinguishing it from cells with non-specific autofluorescence (Fig. S2). Using extracted gDNA of clones 3 and 5, PCR was performed to amplify the gRNA containing region, which was introduced by the CRISPR activation library. Then, the PCR product was employed for TA cloning, followed by colony PCR. By this method, four candidate genes—*PLEKHO2 , MICU*, *MBTPS1*, and *M1AP* were identified (Table 1).

Each candidate gRNA was transfected to the HEK 293T cells using the CRISPR activation system accompanied by p*MYC*-promoter-Dendra2. Because Dendra2 was positive only in the cells transfected with *M1AP* gRNA (Figs. 3A–3D), *M1AP* was considered as the most promising gene that would increase *MYC* transcriptional activity. Then, double-stranded DNA fragments coding the full length of *M1AP* induced into Doxycycline-inducible lentiviral vector (pTRIPZ-*M1AP-GFP*, Fig. 4A). The pTRIPZ-*M1AP-GFP* was introduced into HEK 293T cells. Seventy-two hours after the induction *M1AP* with DOX (1 μg/ml), high *MYC* expression was observed in DOX-induced cells by RT qPCR (*t*-test, $p < 0.05$, Fig. 4B). Furthermore, in the western blots, an increase of *MYC* expression was also observed in DOX-induced cells (Fig. 4C).

Additionally, a dual-luciferase assay conducted with a reporter containing the *MYC* promoter region showed a significant increase of promoter activity in *M1AP*-overexpressing cells at both 48 and 72 h (Figs. 4D and S3). On the other hand, the *MYC* mRNA, which was isolated at the same time points, showed mildly higher in *M1AP*-overexpressing cells at 48 h but not at 72 h (Figs. 4E and Fig. S3 ). Taken together, although *M1AP* was confirmed

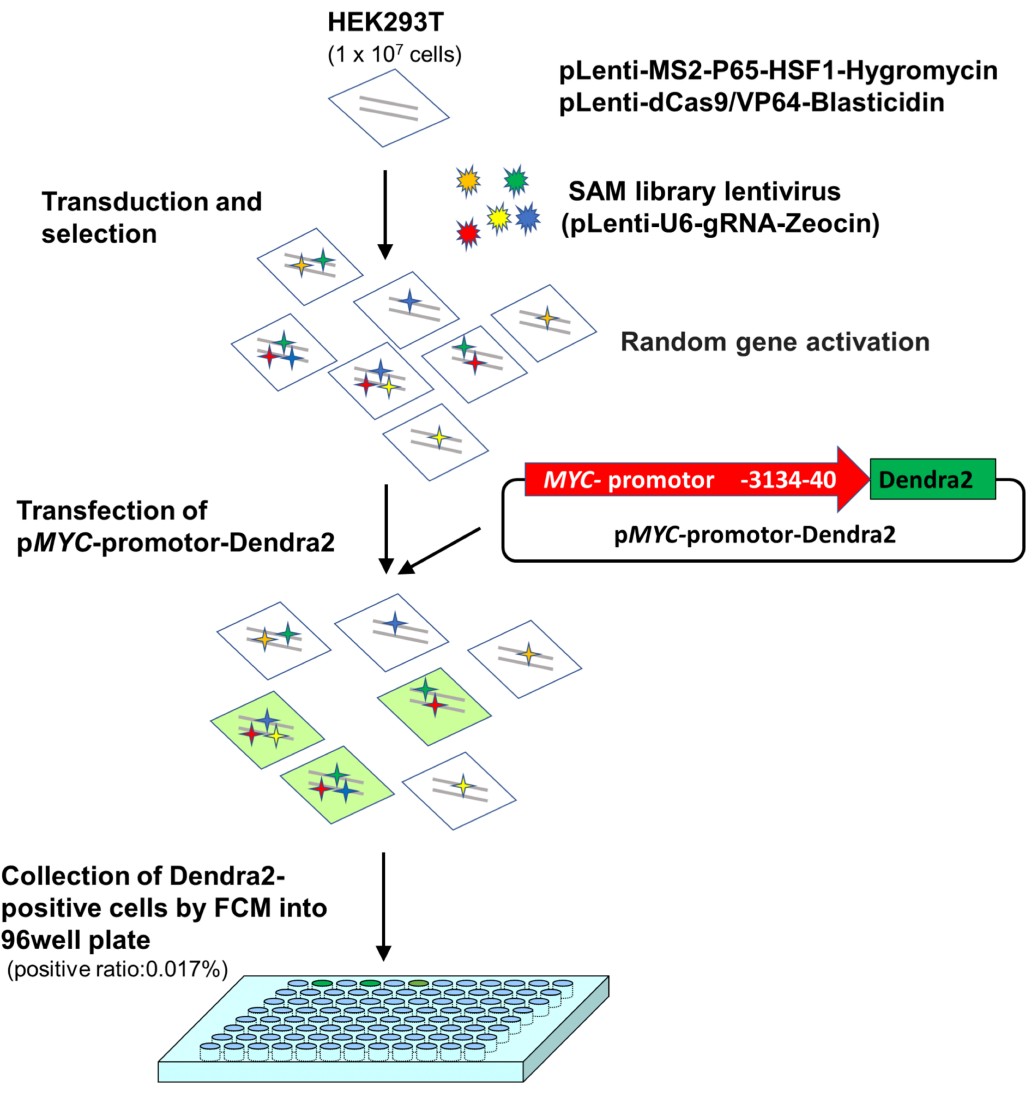

**Figure 1 A scheme for *MYC* transcriptional activator screening with the CRISPR activation library.**

to activate the *MYC* promoter strongly, the *MYC* mRNA itself might be regulated also by other factors.

## Correlation between *MYC* and *M1AP* in the database

According to the GEPIA web server (*Tang et al., 2017*) (http://gepia2.cancer-pku.cn/), high *M1AP* expression is observed in cervical carcinoma, acute myeloid leukemia (AML), ovarian carcinoma, thyroid carcinoma, uterine endometrial carcinoma, and uterine carcinosarcoma when compared to normal tissue (Fig. 4SA). Among these carcinomas, a correlation analysis using cBioPortal revealed a mild positive correlation between *MYC* and *M1AP* only in AML (Fig. 5A, Figs. S4B- S4F). Therefore, we focused on hematopoietic cells, and *M1AP* expression was analyzed using the BloodSpot database. As a result, *M1AP*

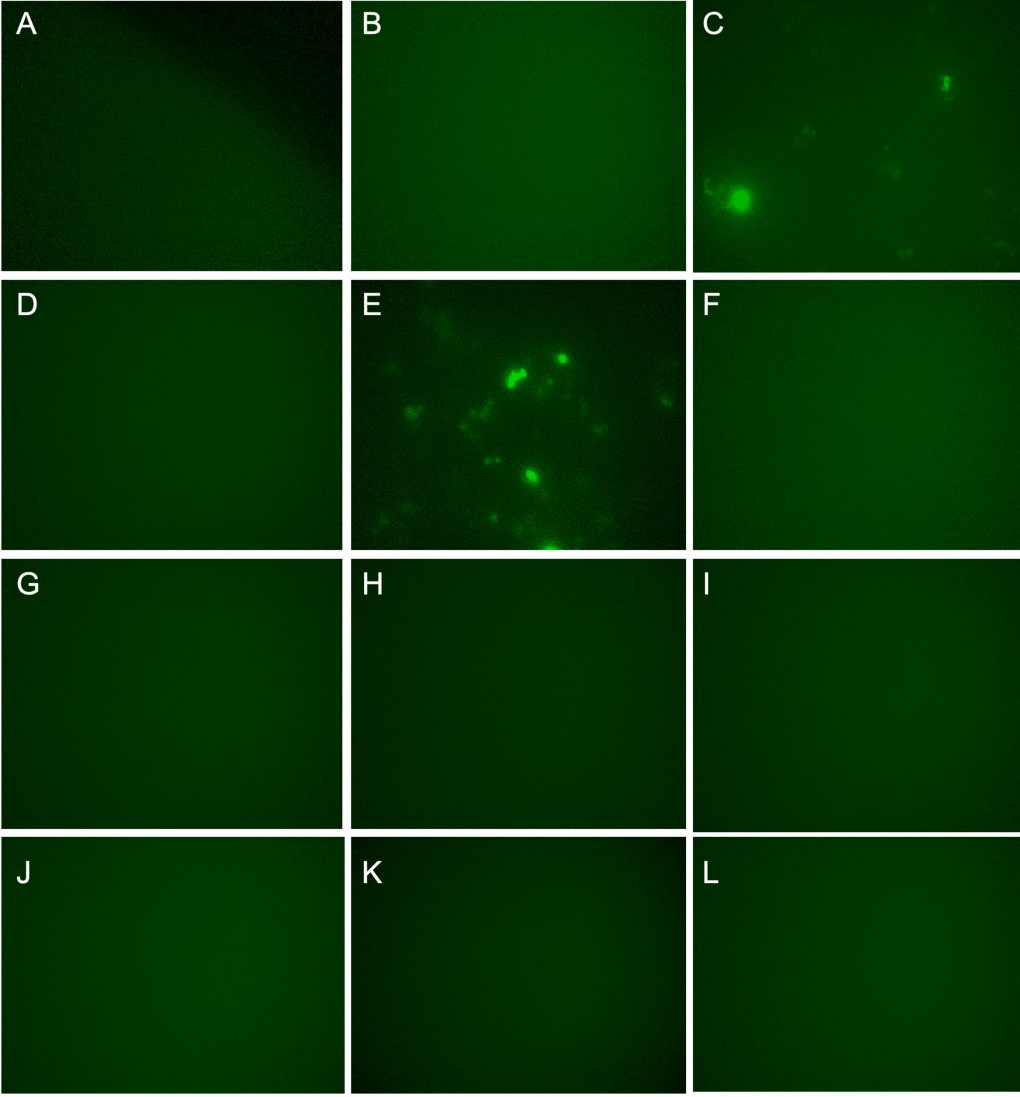

**Figure 2** **Dendra2-positive cells using fluorescence microscope.** (A)–(L) correspond Clone 1-12. p MYC- promoter-Dendra2 was transfected into 12 clones that were successfully cloned after CRISPR activation screening. Clones 3 and 5 showed Dendra2 positivity.

was significantly higher in AML tissue compared to normal bone marrow tissue (Fig. 5B). Interestingly, *M1AP* was highly expressed in AML, especially with t (8;21) (Fig. 5C).

## DISCUSSION

In this study, we created a new reporter vector that incorporates the *MYC* promoter region across the proximal −3.1 kb from the TSS and photoconversion fluorescent protein to evaluate the transcriptional activity of endogenous *MYC*. Ideally, a large region of the *MYC* promoter on Chr.8q24 would be applied to better understand the full *MYC* promoter mechanism. However, even short region on the *MYC* promoter region are still not fully

**Table 1**  List of candidate genes for *MYC* activation.

| Sequence of gRNA | Gene name |
| --- | --- |
| GGGCGGGAAATGGGTGGGGA | *PLEKHO2* |
| TTGTGACACTACTCCAGCCT | *MICU1* |
| AGGATCCCCGAAAAGGAGCA | *MBTPS1* |
| TCTCAGGGTATCTAGGGACT | *M1AP* |

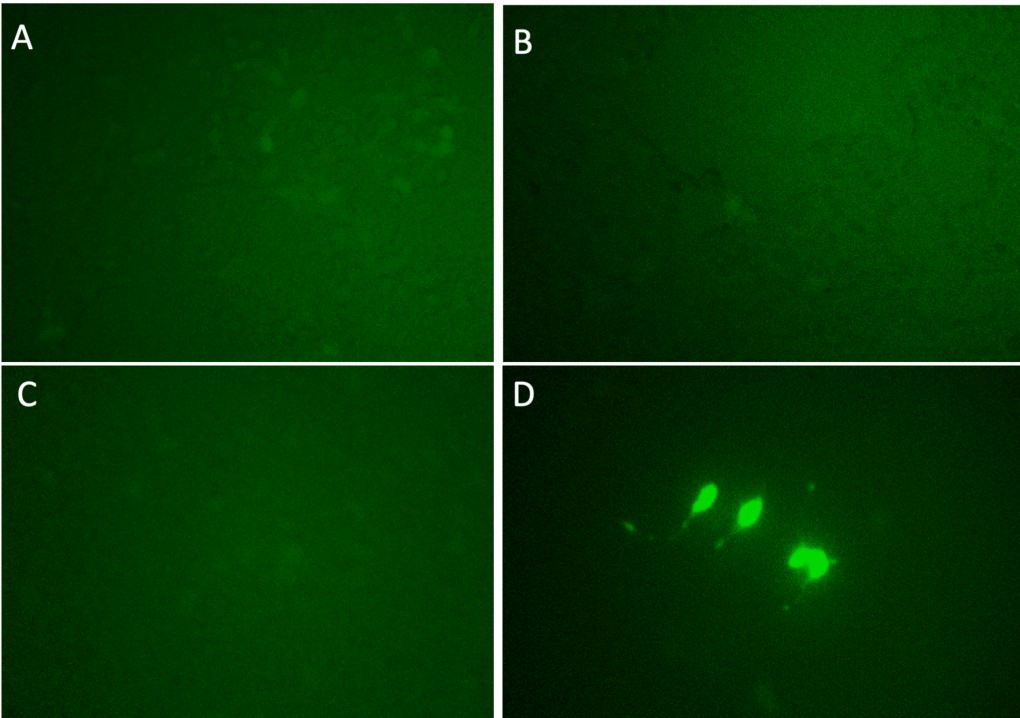

**Figure 3**  **Validation experiments for candidate genes as *MYC* activators.**  (A) *PLEKHO2* (B) *MICU3* (C) *MBTPS1* (D) *M1AP*. With gRNAs of the four nominated genes, p MYC- promoter-Dendra2 was transfected into HEK 293T cells by applying the CRISPR activation system. Only *M1AP*-introduced cells were Dendra2 positive.

understood. In designing this new vector, 3.1 kb of the promoter region was applied because of the limited length of effective DNA insertion into the vector. Future improved experiments on larger region will help to clarify the whole mechanism of *MYC* regulation via the promoter region.

By applying this tool to the CRISPR library, we screened molecules that increase the expression of endogenous *MYC*. According to previous reports, by introducing the CRISPR library into a malignant tumor cell line and carrying out selection with a drug, molecules that cause susceptibility or resistance to that drug can be identified (*Konermann et al., 2015*; *Gilbert et al., 2014*; *Kampmann, 2018*; *Kurata et al., 2018a*; *Kurata et al., 2018b*; *Shalem et al., 2014*; *Wang et al., 2014*; *Koike-Yusa et al., 2014*; *Doench et al., 2016*;

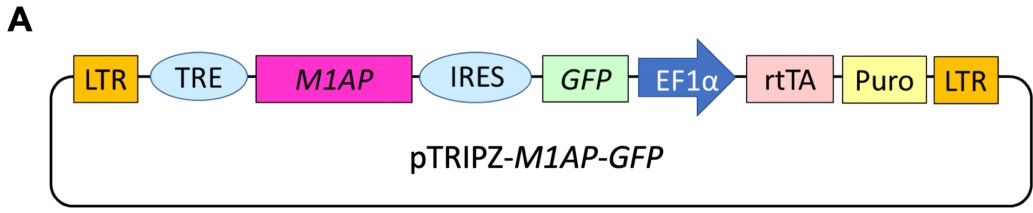

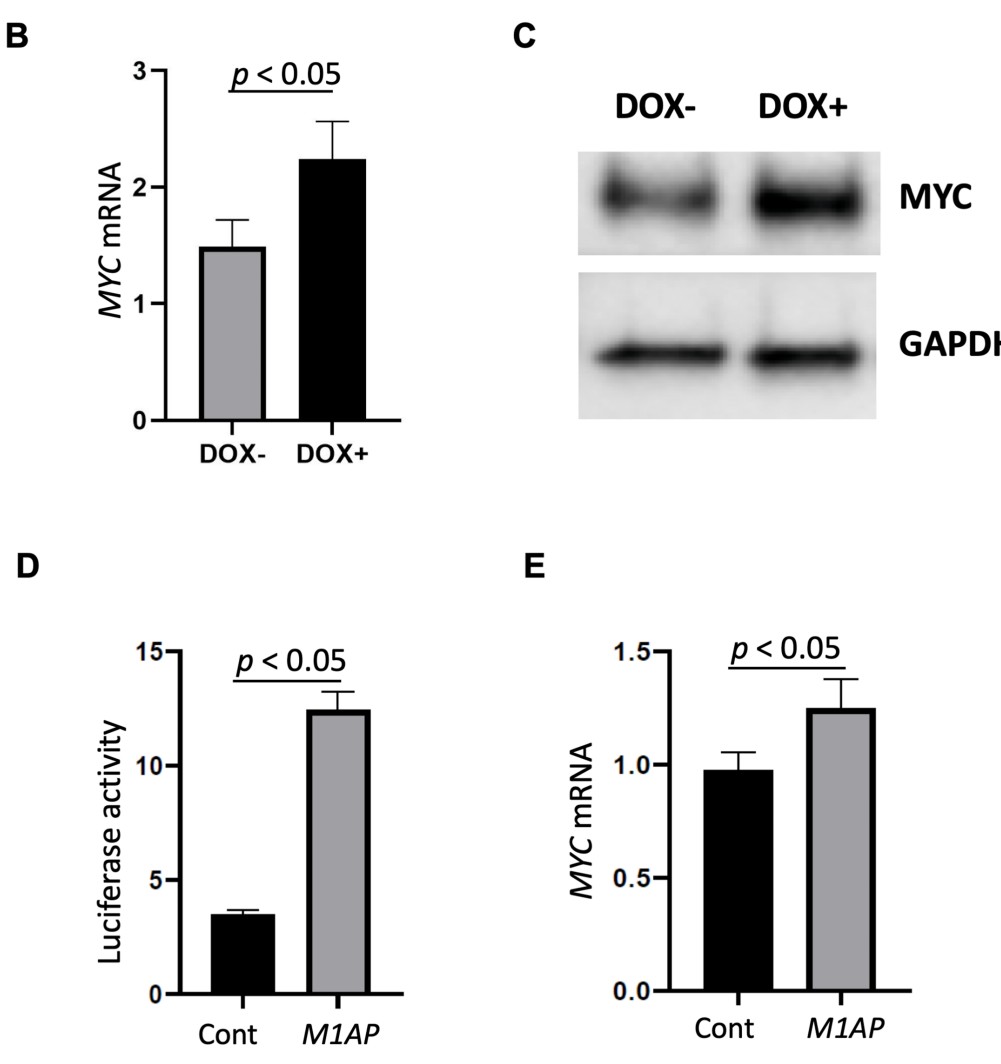

**Figure 4 *MYC* mRNA expression analyzed by real-time quantitative polymerase chain reaction.** (A) Vector map of Dox-inducible *M1AP* expression vector. (B) High *MYC* expression was observed in DOX-induced cells ($t$-test, $p < 0.05$). (C) Western blotting analysis showed increasing of MYC protein in DOX-induced cells. (D) Luciferase activity at 48 hours was significantly higher in *M1AP*-overexpressing cells ($t$-test, $p < 0.05$). (E) *MYC* mRNA expression at 48 hours was mildly higher in *M1AP*-overexpressing cells ($t$-test, $p < 0.05$).

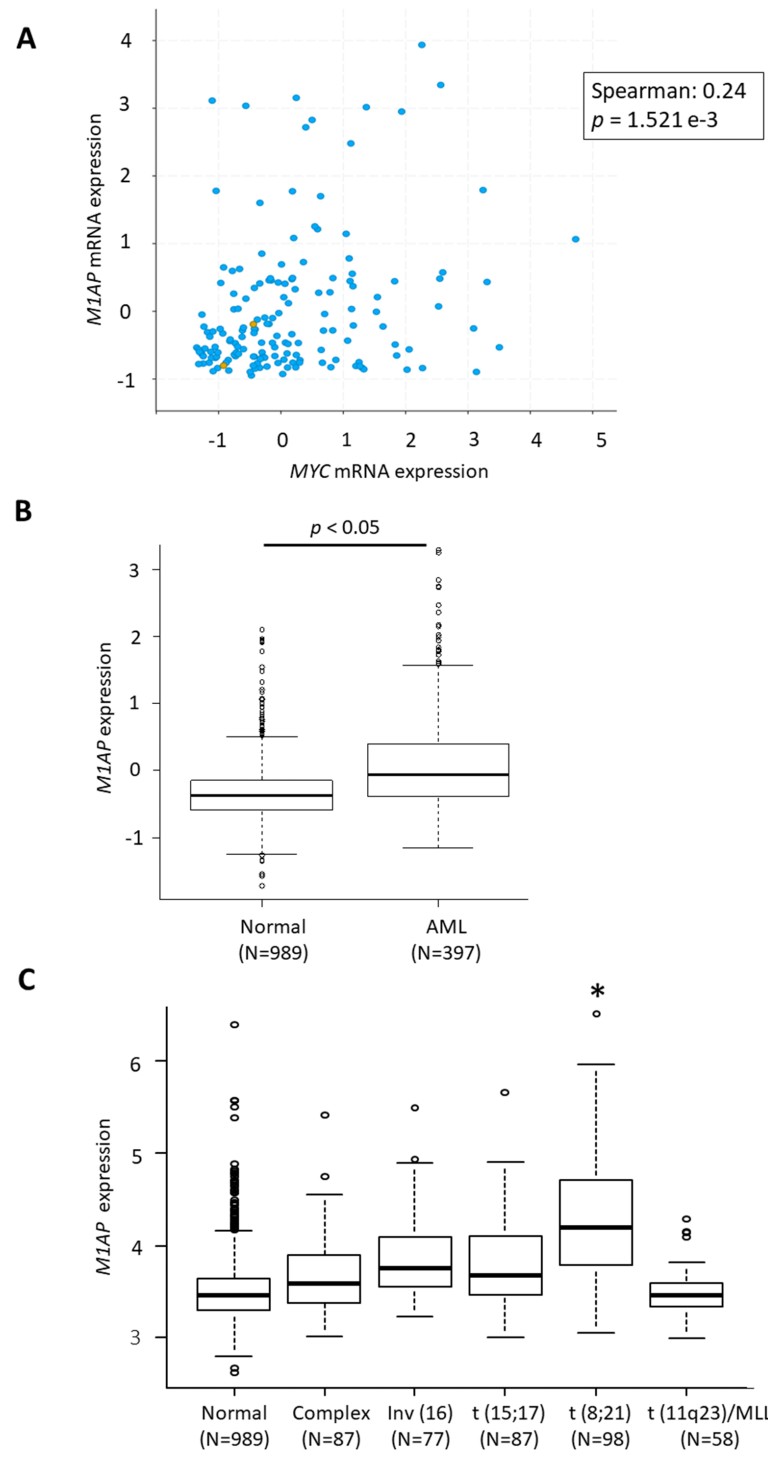

**Figure 5 Correlation between *MYC* and *M1AP* in the database.** (A) In cBioPortal using TCGA data, a correlation analysis between *MYC* and *M1AP* in AML revealed a mild positive correlation. (B) Using the Bloodspot database *M1AP* mRNA expression in acute myeloid leukemia (AML, $N = 989$) was significantly higher than in normal tissue ($N = 397$). (Mann–Whitney U test, $p < 0.05$) (C) The BloodSpot database was used to retrieve gene expression data. *M1AP* expression was higher in each subtype of AML, especially with t (8;21). (Kruskal–Wallis test, *: $p < 0.01$).

*Jain et al., 2016*). However, this is the first specific report of a molecule that activates transcription upstream of *MYC*.

*MYC* is an indispensable factor for cell growth and proliferation in normal tissues, but at the same time, it is one of the most well-known proto-oncogenes, with high amplification in various carcinomas. Moreover, *MYC* is widely involved in tumor development and growth (*Dang, 2012*). Elucidation of factors that regulate the transcriptional activity of *MYC* is crucial for establishing new therapeutic strategies for tumors. Although *MYC* expression is strictly regulated by mRNA and MYC protein, in 1998, *He et al.* reported that inactivation of *APC* located downstream of Wnt signaling pathway results in excessive accumulation of β-catenin and promotes endogenous *MYC* expression by acting on TCF present in the promoter region of *MYC*.

Regarding the research of the regulatory mechanism of *MYC*, *Vo et al. (2018)* transplanted CRISPR/dCas9-VP160 and gRNAs that activated *MYC* transcription in the mouse brain, resulting in the development of medulloblastoma. They concluded that this model is useful for future drug research by suppressing *MYC* expression.

Because this screening was adapted to about 70,290 kinds of gRNAs, initially, multiple molecules were expected to be identified; however, the screening eventually led to the identification of only one molecule, *M1AP*. The reason for this is that, on performing single-cell collection using flow cytometry, to avoid collecting false-positive cells, the cells with the highest fluorescence intensity were collected; as a result, the number of collected cells was extremely small. Furthermore, since the proliferation of the cells after the collection was poor, it is expected that collected cells included cells that were highly brightened due to cell senescence. Moreover, oncogene-induced cell senescence can be one explanation for why the survival rate of collected single cells was very low. Oncogene-induced senescence, which is now a well-known phenomenon, causes cell growth arrest where the oncogene is excessive. This was first reported in *RAS* (*Serrano et al., 1997*), and the same mechanism has been reported for *MYC*, where *MYC* oncogene overexpression results in inactivation of *MYC*, followed by the arrest of cell growth and proliferation (*Wu et al., 2007*). Excessive *MYC* expression due to the CRISPR activation system may have induced cell senescence and caused a low survival rate of sorted cells. We would like to use a next-generation sequencing system and adjust the multiplicity of infection in future screenings.

By newly established experimental line in this study, *M1AP*, which was found as a transcription factor for *MYC*, was proved to function strongly as a *MYC* activator by promoter assay. On the other hand, *MYC* expression was mildly elevated in the *M1AP*-overexpressing cells collected at the same time as the luciferase assay. These results suggest that the *MYC* mRNA is complexly controlled by multiple factors and multiple pathways.

The present study confirmed that this experimental method applying the CRISPR library technology functioned effectively for the identification of the molecule that activates endogenous *MYC*. This method will help elucidate the regulatory mechanism of *MYC* expression, as well as facilitating further drug research against malignant tumors. However, more investigation is required to establish a more accurate and precise screening technique.

## ACKNOWLEDGEMENTS

The authors would like to thank: a member of the stem cell laboratory, Tokyo Medical and Dental University, for cell sorting and her technical assistance; Dr. David Largaespada, Masonic Cancer Center, University of Minnesota, for kindly gifting HEK 293T cells; and Dr. Branden Moriarity, Masonic Cancer Center, University of Minnesota, for plasmids of pT3.5–CAG–DEST.

### Funding
The authors received no funding for this work.

### Competing Interests
The authors declare there are no competing interests.

### Author Contributions

- Akiko Yamamoto performed the experiments, analyzed the data, prepared figures and/or tables, and approved the final draft.
- Morito Kurata conceived and designed the experiments, performed the experiments, analyzed the data, prepared figures and/or tables, and approved the final draft.
- Iichiroh Onishi analyzed the data, authored or reviewed drafts of the paper, and approved the final draft.
- Keisuke Sugita, Miwa Matsumura, Sachiko Ishibashi and Masumi Ikeda performed the experiments, prepared figures and/or tables, and approved the final draft.
- Kouhei Yamamoto and Masanobu Kitagawa analyzed the data, authored or reviewed drafts of the paper, and approved the final draft.

### Data Availability
WB for c-MYC: Kurata, Morito (2020): Western Blot for c-MYC. figshare. Figure. https://doi.org/10.6084/m9.figshare.11991639.v1.

Dendra2 Sorting: Kurata, Morito (2020): Sorting FCS files: CRISPR activation library screening with Dendra2-positive cells. figshare. Dataset. https://doi.org/10.6084/m9.figshare.11423721.v1.

MYC q-PCR: Kurata, Morito; Yamamoto, Akiko (2020): Overexpression vector for M1AP with a doxycycline-inducible vector confirmed that M1AP induced high MYC expression by real-time quantitative PCR. figshare. Dataset. https://doi.org/10.6084/m9.figshare.11423718.v1.

MYC-Luciferase Reporter Assay and Q-PCR for MYC at the same timing with Luc-assay: Kurata, Morito; Yamamoto, Akiko (2020): MYC-Luciferase Reporter Assay and q-PCR for MYC at the same timing with Luc-assay. figshare. Dataset. https://doi.org/10.6084/m9.figshare.11954970.v1.

## Supplemental Information

Supplemental information for this article can be found online at http://dx.doi.org/10.7717/peerj.9046#supplemental-information.

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
