# Peer review of "CRISPR screening identifies M1AP as a new MYC regulator with a promoter-reporter system"

_PeerJ, doi:10.7717/peerj.9046_

## Round 0.1 · original submission · Major Revisions

When evaluating the revisions, please also address a fundamental concern raised that myc regulation spans a 3 MegaBase region on chr8q24. This study have fused only the proximal 3kb to an artificial reporter to screen for regulators. It is crucial to recognize this limitation.

Reviewer 1 ·

Basic reporting

No comment

Experimental design

In this study the authors aim to study the regulators of MYC expression using pooled CRISPR activation screening approach on a reporter line. The approach the authors choose is a valid one, as genome-wide screens provide an unbiased approach to identify novel regulators.

Using a genome-wide screen, the authors uncover one new regulator of MYC expression. The study is a fairly straightforward pooled CRISPR screen but the screen itself has been performed to a poor technical standard. The authors could argue that as long as one regulator is identified, it does not matter how the screen was performed. But this is a missed potential of a technology and these points should be considered when performing such screens in the future.

o The authors do not create a stable reporter line but rather first generate a genome-wide activation library and then transfect the reporter construct into the samples. This is bound to create huge variations. Was the transfection levels controlled? Could it be the case that some cells received multiple copies of the plasmid and other not? Reporter line screens have been reported in a number of studies (at least for CRISPR KO studies, see Breslow et al, 2018, Nature Genetics for example) and it is generally advisable that a stable reporter line is generated before performing pooled screens to avoid variations.

o What was the multiplicity of infection while making the genome-wide libraries?

o The number of cells selected with the phenotype is very very small. Normally from a genome-wide screen, one should expect redundant gRNAs enriched in the selected population. This is not the case in this screen, where the regulator is identified from a single clone.

Validity of the findings

Given the screen has not been performed to a high technical standard, the authors should tighten their study for the validation of M1AP as a regulator of MYC expression.

1.The authors should discuss in what context do they envision this regulation to take place. The authors have performed a very cursory analysis of the expression and only looked at hematopoietic cells. A quick search for expression profile in different cancer types shows that M1AP is up-regulated in a number of different cancer types (THYM,UCEC, OV, CESC (please see attached file). A more detailed study on expression on different cancer types is required.

2. Is there a possibility of knocking out M1AP in a hematopoietic cell line and studying how that affects MYC expression?

3. Based on CRISPR/Cas9- based essentiality screens, M1AP is not essential for any of the 676 cell lines that have been tested up to now (see https://depmap.org/portal/gene/M1AP?tab=overview&characterization=expression). How this does reconcile with its role as a regulator of MYC?

Annotated reviews are not available for download in order to protect the identity of reviewers who chose to remain anonymous.

Reviewer 2 ·

Basic reporting

This paper is clearly written. The introduction and background are reasonable given the premise of the paper.

The paper generates the following kinds of conclusions:
1)A CRISPR activation library was used for the detection of new transcriptional activators of MYC;
2)They found that overexpression of M1AP can promote MYC expression;
3)They have developed a new tool to assess MYC activity and establish screening methods.

Because that the autofluorescence is highly problematic for collecting fluorescent cells in the screening. Using a photoconversion protein, Dendra2 for fluorescent screening is commendable. However, it is not clear how M1AP regulates MYC expression. The association between MYC and M1AP in the database has not been further verified experimentally.

Experimental design

In general the experimental design was clearly written. Some minor changes, additions,modifications would be suggested as follows:

1)pg11 ln:135: Why is the survival efficiency of cells so low, it is necessary to carry out parallel repeated experiments.
2)Fig. 1SB, what the sequences of sgRNAs for HPRT and MYC?

Validity of the findings

Most of the experimental results are reasonable, but there are still some problems.

1)pg13 ln:183-185: For this result, the corresponding experimental process description is unclear.
2)pg14 ln:192-194: There are lack of experimental data to support this result, “more than 23,430 different genes were randomly activated”.Usually,the sgRNA content should be verified in Lentivirus infected cells via NGS analysis.sgRNA representation should be measured at every key step in the process: amplification, viral production, or target-cell transduction.
3)pg15 ln:195: Inaccurate result description. Because the target plasmid was transfected with Lipofectamine3000 reagent, which is not a stably “pMYC-promotor-Dendra2”cell line. And considering that the transfection efficiency may not reach 100%, it is impossible that “pMYC-promotor-Dendra2 was transfected to each cell”.

Additional comments

Interesting paper definitely addresses a need of the scientific community. However, there are some problems with the inaccurate description of the experimental process and results, which need to be carefully modified by the author.

Reviewer 3 ·

Basic reporting

The manuscript entitled “CRISPR screening identifies M1AP as a new MYC 1 regulator with a promoter-reporter system” by Akiko Yamamoto et al. identified a novel MYC regulator called M1AP.
The manuscript is overall well-structured and focused. I think the major conclusions are well supported by experiments. The findings are interesting and will be utilized by the scientific community. I think the manuscript is suitable for the journal PeerJ and can be accepted after minor revision.

Please look for the grammatical errors present throughout the manuscript.
In some places the message that authors want to convey is not clear. For example:
Line 44-45: “The two cells were analyzed to identify containing guide RNAs (gRNAs)”. Please rephrase the sentence for clarity.
Line 109: “The was with Double-stranded DNA…”. Please re-write the sentence.

Experimental design

Experiments are well designed and executed.

Validity of the findings

Finding is novel and interesting

Additional comments

Please check carefully for the missing words in sentences.

---

## Round 0.2 · accepted · Accept

Thanks for the revisions made. All reviewers are in agreement that the paper is acceptable for publication.

Reviewer 1 ·

Basic reporting

No comment

Experimental design

For my concerns on the way the screen was performed, they have now provided sufficient clarifications on the technical reasons on why it was conducted in the particular manner, while being mindful of how it can be improved in the future.

Validity of the findings

The authors have put significant effort in polishing up the paper by providing new analysis and interpretations. I appreciate that the authors have looked more into the context and the manner in which the regulation by M1AP is done and have provided a conclusion that is fitting to the supported results.

Reviewer 2 ·

Basic reporting

As presented, the writing is acceptable for the journal.

Experimental design

As presented, the writing is acceptable for the journal.

Validity of the findings

As presented, the writing is acceptable for the journal.

Additional comments

I would be very glad to re-review the paper in greater depth once it has been edited because the subject is interesting. The Submission has been greatly improved and is worthy of publication.

Reviewer 3 ·

Basic reporting

Authors have incorporated all the necessary suggestions in the revised manuscript. Revision looks satisfactory and has improved the quality of the manuscript.

Experimental design

no comments

Validity of the findings

no comments

Additional comments

The revised manuscript can be considered for publication in PeerJ.